# Unified Model of Shot Noise in the Tunneling Current in Sub-10 nm MOSFETs

**DOI:** 10.3390/nano11102759

**Published:** 2021-10-18

**Authors:** Jonghwan Lee

**Affiliations:** Department of System Semiconductor Engineering, Sangmyung University, Cheonan 31066, Korea; jhlee77@smu.ac.kr

**Keywords:** shot noise, source-to-drain tunneling current, gate tunneling current, sequential tunneling, Fano factor

## Abstract

A single unified analytical model is presented to predict the shot noise for both the source-to-drain (SD) and the gate tunneling current in sub-10 nm MOSFETs with ultrathin oxide. Based on the Landauer formula, the model is constructed from the sequential tunneling flows associated with number fluctuations. This approach provides the analytical formulation of the shot noise as a function of the applied voltages. The model performs well in predicting the Fano factor for shot noise in the SD and gate tunneling currents.

## 1. Introduction

Advances in nanofabrication technology have opened up the possibility of surpassing the ultimate performance limits of modern CMOS devices [1,2]. However, the sub-10 nm channel length in MOSFETs gives rise to tunneling current and associated shot noise through the source-to-drain (SD) potential barrier [3,4,5,6,7,8,9,10,11,12,13,14,15]. With the downscaling of device dimensions, the continuous reduction in gate oxide thickness leads to enormous gate tunneling currents and noise [16,17,18,19,20,21,22,23]. Therefore, the off-state currents, from which the static power dissipated by MOSFETs originates, are enhanced by the direct tunneling current through the SD potential and gate oxide in the subthreshold region and consequently prevent low voltage/power operation [5,6,7]. In addition, the miniaturization of nanoscale CMOS logic devices has made the quantum nature of current flow more pronounced, leading to higher fault rates due to shot noise in the subthreshold region [14]. These factors set fundamental limits on future CMOS technologies, because significant tunneling currents and shot noise are expected under normal operating conditions. The intriguing properties of shot noise, which have been extensively studied in mesoscopic devices, should manifest themselves in sub-10 nm MOSFETs, as the devices enter the ballistic transport regime [9,10,11,12,13,14,15]. However, analytical and highly predictive noise models in ballistic MOSFETs for circuit-level simulation are still lacking [10,11,12]. Although the need for a unified model of shot noise in the SD and gate tunneling current arises frequently in quasi-ballistic MOSFET operation, these currents are typically treated separately for each case [9,10,11,12,13,14,15,21,22,23]. In this work, a single unified analytical expression is presented for the SD and gate tunneling currents and the associated shot noise, emphasizing their similarities and differences. It provides a fully analytical and explicit function of bias conditions and device structural parameters and can be suitably implemented into a circuit simulator. Moreover, the analytical model under a single Landaur transport formula can be helpful to gain easy insights into the shot noise properties of nanoscale MOSFETs in the ballistic regime.

## 2. Tunneling Current Model

For extremely short channel lengths and ultrathin gate oxide thickness, the tunneling current with 2D (*n* = 2) and 3D (*n* = 3) density-of-states (DOS) between two electron reservoirs separated by the energy barrier is given by the Landauer formula [4,24]
(1)$$I_{\alpha }=\frac{2q}{h}{\left(\frac{2\pi m^{*}}{h^{2}}\right)}^{\frac{n-1}{2}}\iint T_{\alpha }\left(E_{s}\right)\left(f_{L}-f_{R}\right)\frac{dE_{t}}{{\left(\pi E_{t}\right)}^{\frac{3-n}{2}}}dE_{s}$$
where q is the charge; h is the Planck’s constant; $m^{*}$ is the effective mass, and n=2 and 3 correspond to source-drain and the channel-gate tunneling, respectively. $f_{L}$ and $f_{R}$ are the Fermi–Dirac distribution functions of the transversal energy $E_{t}$ and longitudinal energy $E_{s}$ at the left and right reservoirs, respectively. For 1D DOS based on carbon nanotubes and Si nanowires, the tunneling current can be expressed as an integral over the longitudinal energy $E_{s}$ [7,24,25]. By an extension of the formulation of (1), the model for p-channel Ge nanowire is modified to take into account phonon and surface roughness scattering in the channel and SD tunneling [25,26]. $T_{\alpha }\left(E\right)$ is the tunneling probability of the energy barrier between source and drain (α=sd) or between channel and gate (α=cg). The tunneling probability is given by the WKB integral for the energy barrier profile $E_{b}\left(s\right)$ as
(2)$$T_{\alpha }\left(E_{s}\right)=\exp\left[-\frac{2\sqrt{2m^{*}}}{\hbar }\int_{s1}^{s2}\sqrt{E_{b}\left(s\right)-E_{s}}ds\right]$$
where, ℏ is the reduced Planck’s constant; $s_{1}$ and $s_{2}$ are the turning points at which $E_{s}$ is equal to $E_{b}\left(s\right)$; and s is in *x*-direction for the channel-gate tunneling and in *y*-direction for source-drain tunneling, respectively, as shown in Figure 1. Using the WKB approximation of (2) for parabolic and trapezoidal barriers of the SD and gate tunneling, respectively, compact models for the tunneling probability can be obtained to derive analytical formulations of the tunneling current and its shot noise [4,7,16,19]. For the source-drain tunneling path, when the channel length L is sufficiently short, the barrier potential $V_{b}\left(y\right)=-E_{b}\left(y\right)/q$ can be approximately represented by a parabolic function of *y* as [5,7]
(3)$$V_{b,sd}\left(y\right)=-A{\left(y-y_{max}\right)}^{2}+\phi_{b,sd}$$
where $V_{b}\left(y\right)$ has a local maximum $\phi_{b,sd}$ at $y=y_{max}$. The boundary conditions are $V_{b}\left(0\right)=V_{bi}$ and $V_{b}\left(L\right)=V_{bi}-V_{ds}$, which yield
(4)$$A=q\left(\phi_{b,sd}-V_{bi}\right)/y_{max}^{2}$$
(5)$$y_{max}=\left(L/V_{ds}\right)\cdot \left(V_{bi}-\phi_{b,sd}\right)\left(1-\sqrt{\frac{V_{bi}-V_{ds}-\phi_{b,sd}}{V_{bi}-\phi_{b,sd}}}\right)$$
where $V_{bi}$ is the built-in potential of the source-channel junction, and $V_{ds}$ is the drain-source voltage. The maximum barrier height $\phi_{b,sd}$ is expressed as a function of $V_{gs}$ and $V_{ds}$
(6)$$\phi_{b,sd}\left(V_{gs},V_{ds}\right)=-\psi_{s}^{QM}\left(V_{gs}\right)-\Delta V_{th}^{DIBL}\left(V_{ds}\right)$$
where $\psi_{s}^{QM}$ is the surface potential calculated by an explicit model [27], and $\Delta V_{th}^{DIBL}$ is the threshold voltage shift due to drain-induced barrier lowering (DIBL) [4]. By substituting (3) into (2) and performing the integral in (2), the source-drain tunneling probability is obtained as
(7)$$T_{sd}\left(E_{y}\right)=\exp\left[-\frac{\pi \sqrt{2m_{Si}}}{\hbar }\frac{y_{max}\left(q\phi_{b,sd}-V_{bi}-E_{y}\right)}{\sqrt{q(\phi_{b,sd}-V_{bi})}}\right]\;$$
where $m_{Si}$ is the electron effective mass along the *y*-direction.

The gate tunneling mechanisms can be primarily divided into Fowler–Nordheim (FN), Poole–Frenkel (PF), and direct tunneling [17,19,23]. For non-stressed devices with a gate oxide thickness below 3 nm, direct tunneling is a dominant mechanism of the gate leakage current [23]. This direct tunneling consists of current components such as channel-gate, S/D-gate, and substrate-gate current. However, for $V_{gs}>0V,\;$the channel-gate tunneling current becomes a main component, since a higher oxide voltage between the channel and the gate causes the current to be proportional to the channel length [17,23]. For the channel-gate tunneling path through the oxide layer, the barrier potential $V_{b}\left(x\right)$ will be $V_{b,cg}\left(x\right)=\phi_{b,cg}-V_{ox}x/t_{ox}$, where $\phi_{b,cg}$ is the barrier height above the conduction band edge at the Si-SiO_2_ interface; $V_{ox}$ is the oxide voltage, and $t_{ox}$ is the oxide thickness. Integration of (2) from x=0 to $x=t_{ox}$ for the gate tunneling probability with $V_{b,cg}\left(x\right)$ gives [16,17]
(8)$$T_{cg}\left(E_{x}\right)=e^{-\frac{4\sqrt{2m_{ox}}t_{ox}}{3\hbar qV_{ox}}\left[{\left(q\phi_{b,cg}-E_{x}\right)}^{\frac{3}{2}}-{\left(q\phi_{b,cg}-E_{x}-qV_{ox}\right)}^{\frac{3}{2}}\right]}$$
where $m_{ox}$ is the electron effective mass in the oxide.

Introducing the ballistic efficiency γ associated with the carrier flows and integrating over $E_{t}$ in (1), the general form of the tunneling currents is given by
(9)$$I_{\alpha }=I_{0}\gamma \int_{0}^{\infty }T_{\alpha }\left(E_{s}\right)\left[F_{\frac{n-3}{2}}\left(\frac{E_{fL}-E_{s}}{k_{b}T}\right)-F_{\frac{n-3}{2}}\left(\frac{E_{fR}-E_{s}}{k_{b}T}\right)\right]dE_{s}\;$$
(10)$$I_{0}=\frac{2q\pi^{n-2}}{h}\;{\left(\frac{2m^{*}k_{b}T}{h^{2}}\right)}^{\frac{n-1}{2}}$$
where $k_{b}$ is the Boltzmann constant; T is the temperature; $E_{fL}$ and $E_{fR}$ are the Fermi-energy level from the ground energy at the left and right reservoirs, respectively, and $F_{\left(n-3\right)/2}\left(u_{0}\right)=\int_{0}^{\infty }u^{\left(n-3\right)/2}/\left(1+e^{u-u_{0}}\right)du$ is the Fermi–Dirac integral. For the source-drain current, (9) gives the total current incorporating both the tunneling current $I_{ds}^{tun}\;\left(0\leq E_{y}\leq q(\phi_{b,sd}-V_{bi}\right))$ in the subthreshold region and the thermionic current $I_{ds}^{the}\;\left(q(\phi_{b,sd}-V_{bi}\right)\leq E_{y}\leq \infty )$ in the inversion region [1,2,3,4,5].

## 3. Shot Noise Model

Shot noise is expressed by introducing the Fano factor Γ defined as $\Gamma =S_{I_{\alpha }}\left(0\right)/2qI_{\alpha }$, $S_{I_{\alpha }}\left(0\right)$ being the spectral density of current fluctuations at low frequency (f≈0). Full shot noise (Γ=1) is expected if the electrons tunneling through the barrier follow Poisson statistics. The Fano factor is suppressed (0<Γ<1) or enhanced (Γ>1), depending on correlations associated with the carrier behavior. The microscopic mechanisms responsible for shot noise have been associated with sequential or coherent tunneling through the potential barrier [28,29]. In the sequential tunneling (ST) model, the interaction of carriers results in other tunneling events due to fluctuations in the potential [16,21,28,29]. In the case of stress-induced leakage currents (SILCs) through the gate oxide, the noise properties are explained by trap-assisted tunneling, a two-step process in which electrons first tunnel from the channel to a trap in the oxide, then from the trap to the gate [12,21,22,23,24,25,26,27,28,29,30]. Analogously, by introducing a two-step tunneling process for the single barrier in both the source-drain and channel-gate direction, we can define four different flow rates, namely, the generation flow rate injected from the left or right ($F_{L}^{+}$ or $F_{R}^{-}$) and the recombination flow rate transmitted to the left or right ($F_{L}^{-}$ or $F_{R}^{+}$), as shown in Figure 1a,b [31,32,33]. The electron transport is governed by the total number of carriers between the left reservoir and the barrier, N. In the presence of a generation-recombination process, the flow rates are modulated by the statistical fluctuation of the state occupancies of the left and right reservoir, leading to a number fluctuation ΔN [21]. Using the Langevin equation for the number fluctuation ΔN and performing a Fourier transform, the Fano factor is obtained as [16,21,28,29]
(11)$$\Gamma =1-\frac{2}{\tau_{R}}\frac{\tau }{1+4\pi^{2}f^{2}\tau^{2}}+\frac{2}{\tau_{R}^{2}}\frac{\tau^{2}}{1+4\pi^{2}f^{2}\tau^{2}}\;$$
(12)$$\frac{1}{\tau }=\frac{1}{\tau_{L}}+\frac{1}{\tau_{R}},\;\;\;\;\;\frac{1}{\tau_{L}}={\frac{dF_{L}^{-}}{dN}|}_{\bar{N}}-{\frac{dF_{L}^{+}}{dN}|}_{\bar{N}},\;\;\;\;\;\frac{1}{\tau_{R}}={\frac{dF_{R}^{+}}{dN}|}_{\bar{N}}\;$$
where $\bar{N}$ is the average of N; and $\tau_{L}$ and $\tau_{R}$ are the characteristic times related to ΔN due to current fluctuations through the left and right reservoirs, respectively. If the flow rate of injected carriers from the left, $F_{L}^{+}$, in the single barrier is independent of N, then the characteristic times $\tau_{L}$ and $\tau_{R}$ given in (12) are represented by the derivatives of the carrier flow to the left, $F_{L}^{-}$, and to the right, $F_{R}^{+}$, respectively. By extending the Landauer Formula (1), $F_{L}^{-}$ and $F_{R}^{+}$ take the forms
(13)$$F_{L}^{-}=\frac{2}{h}{\left(\frac{2\pi m^{*}}{h^{2}}\right)}^{\frac{n-1}{2}}\gamma^{2}\iint (1-T_{\alpha })f_{L}\left(1-f_{L}\right)\frac{dE_{t}}{{\left(\pi E_{t}\right)}^{\frac{3-n}{2}}}dE_{s}\;$$
(14)$$F_{R}^{+}=\frac{2}{h}{\left(\frac{2\pi m^{*}}{h^{2}}\right)}^{\frac{n-1}{2}}\gamma \iint T_{\alpha }f_{L}\left(1-f_{R}\right)\frac{dE_{t}}{{\left(\pi E_{t}\right)}^{\frac{3-n}{2}}}dE_{s}\;\;$$

In order to calculate the flow rate of carriers for 1D DOS, (13) and (14) can be replaced by integration over the longitudinal energy $E_{s}$ [11,24]. Taking the derivatives of (13) and (14) with respect to N leads to the expression (i = 1 for *L* and i  = 2 for *R*)
(15)$$\frac{1}{\tau_{L,R}}={\left(-1\right)}^{i}\frac{2}{h}{\left(\frac{2\pi m^{*}}{h^{2}}\right)}^{\frac{n-1}{2}}\gamma^{\frac{2}{i}}\iint \frac{dT_{\alpha }}{dN}f_{L}\left(1-f_{L,R}\right)\frac{dE_{t}}{{\left(\pi E_{t}\right)}^{\frac{3-n}{2}}}dE_{s}\;$$
(16)$$\frac{dT_{\alpha }}{dN}=\frac{q\partial T_{\alpha }}{\partial \phi_{b,\alpha }}\;\frac{\partial \phi_{b,\alpha }}{\partial N}+\frac{\partial T_{\alpha }}{\partial V_{\alpha }}\;\frac{\partial V_{\alpha }}{\partial N}\;$$

For source-drain tunneling, the source-drain voltage is approximately independent of N, and therefore$\;\partial V_{\alpha }/\partial N\approx 0$. By differentiating (7) with respect to $\phi_{b,sd}$ and combining all electrostatic effects in a geometrical gate capacitance $C_{g}$, the derivatives of the first term in (16) become
(17)$$\frac{q\partial T_{sd}}{\partial \phi_{b,sd}}=-\frac{q\pi \sqrt{2m_{Si}}y_{max}}{\hbar }\frac{1-\frac{q\phi_{b,sd}-E_{y}}{2q(\phi_{b,sd}-V_{bi})}}{\sqrt{q(\phi_{b,sd}-V_{bi})}}\cdot T_{sd}\;$$
(18)$$\frac{\partial \phi_{b,sd}}{\partial N}=\frac{q}{C_{g}+C_{Qs}+C_{Qd}},\;\;\;\;\;\;C_{Qs,d}=q^{2}\int_{0}^{\infty }D\left(E\right)\left(-\frac{\partial f_{s,d}}{\partial E_{fs,d}}\right)dE\;$$
where $C_{g}=C_{ox}C_{Qg}/\left(C_{ox}+C_{Qg}\right)$; $C_{ox}$ is the oxide capacitance; $C_{Qg}$, $C_{Qs}$, and $C_{Qd}$ are the quantum capacitances of the gate, source, and drain, respectively, and D(E) is the 2D DOS in the channel [10,30]. In the case of channel-gate tunneling through the oxide layer, the derivatives are obtained from (8) and (16) as
(19)$$\frac{q\partial T_{cg}}{\partial \phi_{b,cg}}=-\frac{2\sqrt{2m_{ox}}t_{ox}}{\hbar V_{ox}}\left(a^{\frac{1}{2}}-b^{\frac{1}{2}}\right)\cdot T_{cg}$$
(20)$$\frac{\partial T_{cg}}{\partial V_{ox}}=-\frac{4\sqrt{2m_{ox}}t_{ox}}{3\hbar qV_{ox}^{2}}\left[b^{\frac{1}{2}}\left(a+\frac{1}{2}qV_{ox}\right)-a^{\frac{3}{2}}\right]\cdot T_{cg}$$
where $\partial \phi_{b,cg}/\partial N\approx q/C_{Qg}$, $\partial V_{ox}/\partial N=-q/C_{Qg}$, $a=q\phi_{b,cg}-E_{x}$, and $b=q\phi_{b,cg}-E_{x}-qV_{ox}$ [16]. Note that the set of (11), (12), and (15)–(20) provide the analytical formulation of the shot noise as a function of the applied voltages.

In the coherent tunneling (CT) model, there is no electron scattering during electron transmission through the barrier and the transport is governed by the total transparency of the barrier [14,29]. In the zero frequency limit, the shot noise is given by
(21)$$S_{I_{\alpha }}=2\frac{2q}{h}{\left(\frac{2\pi m^{*}}{h^{2}}\right)}^{\frac{n-1}{2}}\iint \{T_{\alpha }\left[f_{L}\left(1-f_{L}\right)+f_{R}\left(1-f_{R}\right)\right]+T_{\alpha }\left(1-T_{\alpha }\right){\left(f_{L}-f_{R}\right)}^{2}\}\frac{dE_{t}}{{\left(\pi E_{t}\right)}^{\frac{3-n}{2}}}dE_{s}$$
where the first and second term describe the injection noise and the partition noise of coherent tunneling, respectively.

## 4. Results and Discussion

For the calculation of the current and noise, the Fermi-energy of the gate $E_{fg}$ must be numerically solved using the relation [34]
(22)$$e^{\frac{N}{C_{ox}V_{th}}}\left(e^{\frac{N}{C_{ox}V_{th}}}-1\right)=e^{\frac{V_{gs}-V_{T}}{V_{th}}}$$
(23)$$N=\frac{C_{Qg}V_{th}}{q}\ln\left(1+e^{\frac{E_{fg}}{k_{b}T}}\right)$$
where $V_{th}=k_{b}T/q$ is the thermal voltage, and $V_{T}$ is the threshold voltage. The Fermi energy of the source $E_{fs}$ for the entire on-state and subthreshold behavior is given by [34]
(24)$$\;E_{fs}=\ln\left[\sqrt{{\left(1+e^{v_{d}}\right)}^{2}+4e^{v_{d}}\left(e^{2N/N_{Qg}}-1\right)}-\left(1+e^{v_{d}}\right)\right]-\ln\left(2\right)$$
with $v_{d}=qV_{ds}/k_{b}T$ and $N_{Qg}=C_{Qg}V_{th}/q$. Figure 2a,b show the calculated SD and gate current, respectively, as functions of $V_{gs}$ for different channel lengths and oxide thicknesses.

The simulation for an n^+^ poly-Si nMOSFET with nitrided oxide was performed using (9) and (10) with $\phi_{b,cg}\;$=2.6 eV $t_{ox,eq}\;$=1.5/2.0 nm, $m_{ox}=0.4m_{0}$,$\;m_{Si}=0.19m_{0}$, $V_{T}=0.1\;V$, T=300 K, γ=0.5 for $I_{sd}$, and γ=0.9 for $I_{cg}$. In the case of high-κ dielectric/metal gate stack, the equivalent oxide thickness $t_{ox,eq}$ should be carefully estimated by the capacitance measurement, because a replacement of gate stack alters the electrical thickness of oxide [35]. For the SD current in the subthreshold region, the tunneling contribution becomes dominant, and thermionic contribution is reduced owing to the high channel barrier.

Figure 3a,b show the comparisons between ST and CT model of the Fano factor for SD and gate tunneling shot noise, respectively, as functions of $V_{gs}$. As can be seen from Figure 3a, in the subthreshold region, the noise in the SD current is enhanced (Γ>1), owing to the positive correlation between tunneling into the channel caused by the interplay between the DOS and electrostatics [11,14]. This is consistent with the results obtained by statistical Monte Carlo simulations [11,36], quantum-mechanical injection model [37,38], and Fermi statistics [39]. Actually, the noise enhancement in the subthreshold region is due to the fact that at low source-drain voltage ($V_{ds}=0.1\;V$), the current is small, while the spectral density of shot noise $S_{I_{d}}\left(0\right)$ tends to a finite value, because of the thermal noise contribution [14].

The results in Figure 3a point out the difference for 2D and 1D DOS when the sequential model is used. The Fano factor for 1D DOS is calculated by using the modified formulations of the SD current of (1), the flow rate of carriers of (13) and (14), and the quantum capacitances of (18). When the dimensionality of DOS is reduced, only few electrons take part in transport so that SD current fluctuations normalized to the SD current can heavily affect the noise behavior [40,41]. For moderately high voltages in the inversion region, $f_{R}$ is so low that injection noise from the left reservoir is dominant, leading to noise suppression (Γ<1) [14,40,41].

As shown in Figure 3b, for verification of the shot noise model for the gate tunneling current, the simulation results are compared with the measured Fano factor for an n+ poly-Si nMOSFET as a function of $V_{gs}$ [21]. It can be seen that large enhancement in gate shot noise at low voltages is in good agreement with the experimental results. This is due to the fact that both of the inverse time constants of the ST model are nearly equal in magnitude, but of opposite signs [16,28,42,43]. As $V_{gs}$ increases, the tunneling probability becomes higher, and noise following a partition process appears [14,44].

## 5. Conclusions

The general expressions of current and noise are found to be well-suited for computation in a unified manner of the Fano factor of both SD and gate shot noise in sub-10 nm MOSFETs. The sequential tunneling model associated with the number fluctuations predicts enhancement of the shot noise in the SD tunneling current in the subthreshold regime, thus extending the validity of the model to the enhancement of shot noise in the gate tunneling current at low voltages. The resulting analytical compact models may be considered suitable in circuit simulators for circuit-level simulation.

## Figures and Tables

**Figure 1 nanomaterials-11-02759-f001:**
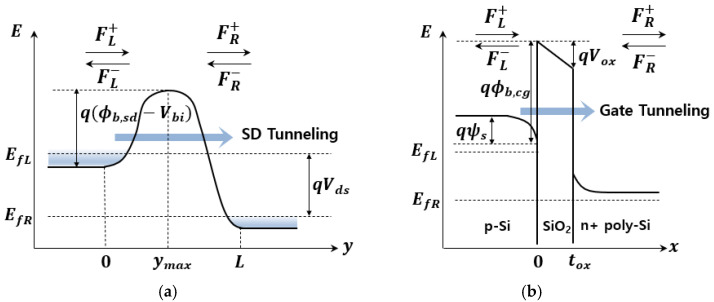
Representation of energy barrier profile (**a**) along the *y*-axis in the source-drain direction and (**b**) along the *x*-axis in the channel-gate direction.

**Figure 2 nanomaterials-11-02759-f002:**
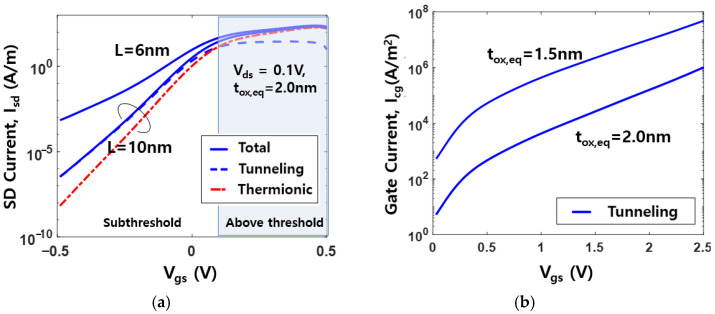
(**a**) Calculated SD current versus $V_{gs}$
for $V_{ds}=0.1V$ (**b**) calculated gate tunneling current versus $V_{gs}$ for different $t_{ox,eq}$.

**Figure 3 nanomaterials-11-02759-f003:**
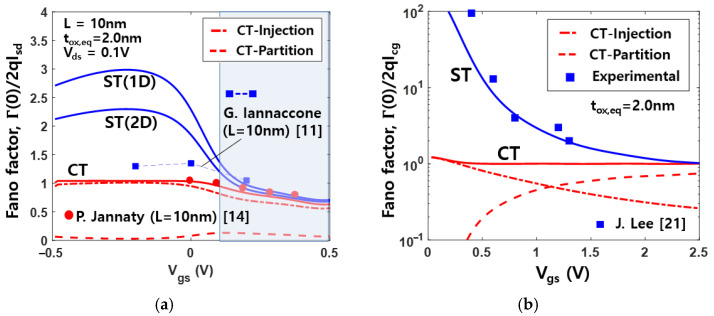
Comparison between ST model and CT model of Fano factor as a function of $V_{gs}$
for (**a**) SD current noise and (**b**) gate tunneling current noise.

## Data Availability

The data presented in this study are available on request from the corresponding author.

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
