# Peer review of "Unified Model of Shot Noise in the Tunneling Current in Sub-10 nm MOSFETs"

_nanomaterials, 2021, doi:10.3390/nano11102759_

Round 1
Reviewer 1 Report
The paper deals with the interesting issue of a compact model for shot noise in nanoscaled devices. The Authors address the issue of a unified analytical model accounting for the source-drain and gate tunneling. The proposed equations for the two contributions are taken from the literature and they are simply written by defining an ad-hoc parameter n in order to find one of the two terms by fixing n=2 or 3. Starting from the current formula, they define the Fano factor for the shot noise model by using again approaches already available in the literature. What is not clear is the advantage of using such compact unified analytical model with respect to two distinct ones, by simply fixing n= 2 or 3 in their formulations. The latter aspect should be the novelty proposed by this work and needs a further clarification for the readers, explicitly showing the advantages of the compact unified model. Moreover, the results are limited to a single device, taken as reference, but the advantage of the proposed model would require to adopt it for a much more extended analysis, by addressing different geometries and leading to interesting outcomes in the predictions of the Fano factor as well. Finally, any additional data already available in the literature which might confirm and validate the model would be necessary.
Reviewer 2 Report
This work by Lee presents an analytical investigation into the shot noise of sub 10 nm transistors considering both SD and gate tunneling current, which is an important issue of the noise spectra of short channel devices. The model is built bottom-up from the fundamental Landauer formula and the sequential tunneling associated with the number fluctuations. The text is well written and the results are clearly presented. However, the content of this manuscript could be further enriched to attract broader readers in the field of semiconductor physics.
Here are some comments and suggestions that might help to further improve the quality of the manuscript.
- I strongly suggest the author relate this research to the state of art CMOS technologies. Short channel CMOS transistor with a channel length of literally less than 10 nm is a challenge. Multibridge channel fets, 2D material fets, and carbon nanotube fets are promising to achieve sub-10 nm channel length. Therefore the channel materials for sub 10 nm transistors could be Si-Ge alloy, 2D materials, or carbon nanotube, rather than merely pure silicon.
- Also, the polySi gate is quite a traditional technology. The author has proved that the model in this work can well fit the experimental data, which is a good thing. The author can further use this model, if possible, to predict the behavior of metal high-k gate, which is used in more advanced CMOS technologies.
- I suggest the author separately list "source-to-drain tunneling current" and "gate tunneling current " in the keywords, which might help to hit more searches from the internet and receive more citations.
Round 2
Reviewer 1 Report
I thank the Authors for the improvements in the manuscript. I still would require some additional results as I mentioned in the first revision. I would suggest to add figures showing the model used for the simulation of different geometries and different materials chosen among those cited in the revised manuscript.
